# COVID-19 among Health Workers in Germany and Malaysia

**DOI:** 10.3390/ijerph17134881

**Published:** 2020-07-07

**Authors:** Albert Nienhaus, Rozita Hod

**Affiliations:** 1Competence Centre for Epidemiology and Health Services Research for Healthcare Professionals (CVcare), University Medical Centre Hamburg-Eppendorf (UKE), 20459 Hamburg, Germany; 2Department of Occupational Medicine, Hazardous Substances and Public Health, Institution for Statutory Accident Insurance and Prevention in the Health and Welfare Services (Berufsgenossenschaft für Gesundheitsdienst und Wohlfahrtspflege—BGW), 22089 Hamburg, Germany; 3Department of Community Health, Faculty of Medicine, National University of Malaysia Jln Yaacob Latif, Bandar Tun Razak, Cheras, Kuala Lumpur 56 000, Malaysia; rozita.hod@ppukm.ukm.edu.my

**Keywords:** COVID-19 pandemic, SARS-CoV-2, mortality, health Worker

## Abstract

We report on the suspected case reports filed for SARS-CoV-2 infections and COVID-19 illnesses among health and social welfare workers in Germany. In addition, we report about COVID-19 in health workers in Malaysia. Claims for occupational diseases caused by SARS-CoV-2 are recorded separately in a database of the Statutory Accident Insurance and Prevention in the Health and Welfare Services (BGW). This database is analyzed according to its content as of May 22, 2020. In addition, the notifiable cases of SARS-CoV-2 infections from personnel in medical institutions (e.g., clinics and doctor’s office) and social welfare institutions (e.g., nursing homes, shelters and refugee camps) following the German Infection Protection Act are analyzed. The report from Malaysia is based on personal experience and publications of the government. In Germany at present, 4398 suspected case reports for the diagnosis of SARS-CoV-2 infections among health and social workers have been filed. This figure is four times the number of all reported infections normally received per year. The majority of claims, regardless of being a confirmed infection, concerned nurses (n = 6927, 63.9%). The mortality rate for workers infected with SARS-CoV-2 is 0.2% to 0.5%. Doctors are affected by severe illness more frequently than other occupational groups (8.1% vs. 4.1%). In Malaysia, work-related infection of health workers (HW) occurred mainly when COVID-19 was not suspected in patients and no adequate personal protective equipment (PPE) was worn. Although knowledge on the spread of SARS-CoV-2 infections among workers remains limited, the impact appears to be substantial. This is supported by the mortality rate among infected workers. Occupational health check-ups carried out at the present time should be systematically analyzed in order to gain more information on the epidemiology of COVID-19 among HW. Since the supply and use of PPE improved, the infection risk of HW in Malaysia seems to have decreased.

## 1. Introduction

The novel coronavirus was first discovered in December 2019 in China. The virus was most likely transferred to humans from wild animals (e.g., bats) at a market in Wuhan. This is corroborated by the observation that the first documented occupational groups at risk were persons working in seafood and wet animal wholesale markets in Wuhan [1]. The first cases in Germany were reported in January 2020. From mid-March to early April, the Robert Koch Institute (RKI) reported as many as 6000 new cases per day. At the time of writing, the number of new cases reported per day is below 500 [2].

Due to the similarity of the new virus from the coronavirus family to the viruses that caused Severe Acute Respiratory Syndrome (SARS) and Middle East Respiratory Syndrome (MERS), it has been named SARS coronavirus 2, abbreviated to SARS-CoV-2, and the respiratory disease it causes has been named COVID-19 (coronavirus disease 2019). Similarly to the case of the influenza A (H1N1) pandemic in 2009, health workers (HW) were particularly affected during the SARS outbreaks in 2002 and 2003, as well as the MERS outbreaks since 2012 [3,4,5,6,7,8,9]. It is safe to assume that with the COVID-19 pandemic frontline, HW are vulnerable particularly because of the shortage of adequate personal protective equipment (PPE) at the beginning of the pandemic [10].

However, official figures on the risks to HW remain limited. The WHO estimates that as of April 8, 2020, over 20,000 health workers in 52 countries had contracted COVID-19 [11]. In Italy alone, there had been 15,314 cases of SARS-CoV-2 infections among HW by April 10, 2020. This corresponded to 11% of all known infections in Italy [12]. In China, HW accounted for a particularly high proportion of cases at the beginning of the epidemic [13]. The improvement of protective measures and the improved use of PPE appear to have substantially reduced the risk over time [14].

In Germany, very few figures have been published to date on the occurrence of infections among HW. Two head physicians from North Rhine-Westphalia caused a stir when they tested positive for SARS-CoV-2 after a ski trip and had already been in contact with patients and colleagues after returning from the vacation (personal communication). Similar cases from Austria are also known. In Malaysia, a wedding was the cause of infection for several HW, as we will learn from the report from Malaysia below.

Along with the risk of infection, the COVID-19 pandemic is leading to further sources of stress for health workers. Overfilled intensive care wards, overfilled emergency wards, long working hours to compensate for absent colleagues who are ill or quarantined, wearing PPE for sustained periods of time, isolation of patients and particularly of residents in old-age and nursing facilities, additional administrative tasks due to reporting obligations, task force meetings, and contact tracings, restricted contact with colleagues, and the fear of transmitting the disease to their own families are factors that can lead to exhaustion and excessive psychological stress for workers [15,16]. Sleep disturbance and even suicidal thoughts can be caused by exposure to COVID-19 patients and the associated increased workload and worries about own safety and health [17,18]. These aspects of the pandemic should also be taken into consideration, in addition to protection against infection, and corresponding support services should be developed. For this reason, a WHO working group has called attention to the fact that the pandemic cannot be allowed to lead to worsened working conditions or failures to comply with occupational safety standards [19].

Neither H1N1 nor SARS or MERS led to a significant increase in the number of infections reported as occupational diseases in Germany [20]. The experiences in China and Italy made it clear that the COVID-19 pandemic would not end quickly and that severe progressions of the disease up to the point of death are likely [12,13]. Therefore, the accident insurance provider for the private health and welfare sector in Germany, the BGW, has established its own system for tracking SARS-CoV-2 cases among its insured persons. In addition to the standardized entry of occupational disease reports in the routine database (BK-DOK), all reported cases of SARS-CoV-2 infections are recorded separately in order to facilitate ongoing analysis.

As this article is part of the special issue of the conference proceedings of the 11th International Joint Conference on Occupational Health for Health Workers in Hamburg (OHHW2019), we asked participants to provide reports of the COVID-19 experience for HW in their country. Malaysia responded positively to this request. 

## 2. Objectives

In this descriptive study, we report the cases of SARS-CoV-2 infections and COVID-19 illnesses in HW in Germany. In addition, a report on the COVID-19 situation for HW in Malaysia is given. 

## 3. Methods

In accordance with the Infection Protection Law (IfSG), the RKI receives information on occupation in a facility relevant for infection control for reported COVID-19 cases. A distinction is made between (A) staff of hospitals, outpatient clinics and practices, dialysis clinics or outpatient nursing services, and (B) staff of facilities for the care of older, disabled, or other persons in need of care, homeless shelters, community facilities for asylum seekers, repatriates and refugees, as well as other mass accommodation and prisons. Local public health institutions, responsible for contact tracings, transmit this information to the RKI. 

Since information on occupations in these facilities is missing in 29% of cases, the proportion of cases working in these facilities published by RKI should be considered minimum values [2]. Among the COVID-19 cases reported as working in the above-mentioned facilities, the proportion of cases that actually acquired their infection in these settings is unknown.

Suspected case reports of occupational diseases are recorded on a regular basis in accident insurance providers’ BK-DOK in a standardized manner. However, the BK-DOK does not allow new infectious diseases to be recorded separately because no numerical codes have been reserved for this. For this reason, all reports to BGW that pertain to SARS-CoV-2 or COVID-19 are systematically recorded in a separate documentation system by the district offices that received them. Along with profession and sector, the system also records reporting obligations, whether a test has been carried out, the result of the test, the progression of the illness, recovery or death, and whether a claim was confirmed or not as occupational disease.

Physicians are obligated to report suspected cases of occupational disease either to federal state-level trade physicians (*Landesgewerbearzt*) or to the responsible accident insurance provider. The obligation to review the responsibility for this lies with the accident insurance provider, however, and not with physicians. They review whether they are responsible for the report/affected person and forward the report to the responsible accident insurance provider if necessary. The same process occurs for reports to federal state-level trade physicians. They also forward the reports to the responsible accident insurance provider.

For this reason, a COVID-19 illness that was presumably occupational in nature must be reported to the accident insurance provider or the federal state-level trade office (*Landesgewerbeamt*). This does not replace or affect the reporting obligation to the public health office pursuant to the Infection Protection Law. However, it must be mentioned as a caveat that occupational disease (OD) 3101, under which COVID-19 falls, only applies for four groups: workers employed in the areas of (1) healthcare, (2) welfare, (3) laboratories, and (4) workers with activities subject to an elevated risk of infection comparable to that for healthcare [21]. In all other situations, a review is carried out in regard to whether the case constitutes an occupational accident.

The reporting obligation applies as of the time when the illness was diagnosed or when there is a reasonable suspicion that a certain illness exists. For COVID-19, the latter applies following contact with infectious patients or materials when the typical symptoms of COVID-19 appear within the appropriate temporal context: cough, fever, disruption of the senses of smell and taste, etc. Corresponding reports can establish eligibility for benefits in accordance with the Occupational Disease Regulation. Reporting mere contact without indications of a corresponding illness is not mandatory and typically does not establish eligibility for benefits.

It is recorded whether a test has been carried out and what the result of the test was (positive/negative). The type of test carried out is not recorded. Initially, only oral-nasal swabs and Polymerase Chain Reaction (PCR) tests were available. Antibody tests are now available as well. The data do not allow for a differentiation to be made here. The progression of an illness is classified as severe if hospitalization was necessary.

The data for the individual reports are recorded when the reports are received, and the BGW administrators contact the insured persons as quickly as possible in order to research missing information and clarify any need for support as early as possible.

Rozita Hod compiled the report from Malaysia based on personal experience and documents published by the government [22,23,24]. 

## 4. Results

### 4.1. SARS-CoV-2 Infections in HW in Germany

Until May 25, 2020, 12,393 cases with a SARS-CoV-2 infection have been notified among staff working in medical facilities as defined by Section 23 of the Infection Protection Law (Table 1). Among the cases reported as working in medical facilities, 73% were female and 27% male. The median age was 41 years (no table) and 20 people died, giving rise to a mortality rate of 0.2%. Among the staff of various care facilities (Section 36 of the Infection Protection Law), 8935 cases are reported and 46 (0.5%) died. The high number of cases is consistent with numerous reported outbreaks, especially in nursing homes. The proportion of recovered patients in the medical facility (96.3%) and the care facility staff group (94.3%) were somewhat higher than for all reported cases in Germany (90.3%). The proportion of patients who died, 0.2% or 0.5%, was lower than for all reported cases (4.1%). The same is true for the hospitalization rate (4.6% or 4.3% versus 17.8%). Taking both groups together, staff from medical facilities or from care facilities represent 11.9% of all the 178,570 SARS-CoV-2 cases reported to the RKI in Germany so far.

As of May 22, 2020, 12,038 cases had been reported to the BGW (Table 2). Of these, 4398 (36.5%) were classified as subject to mandatory reporting. Tests were specified for 10,835 reports. Of these tests, 3690 (34.1%) were positive. Assessments of the progression of the illness are available for 3038 (82.3%) of the cases with a positive test result. Of these cases, 2570 (84.6%) had mild and 151 (5.0%) had severe symptoms. Eleven (0.4%) insured persons died. The decision whether a case was confirmed as OD or not was reached for 3067 (83.1%) cases. The confirmation rate is 71.5%.

The majority of reported cases with a known test result pertain to clinics (74.9%) (Table 3). However, only 23.1% of the affected persons in this section tested positive. 1762 reports are from inpatient or outpatient care (16.3%). Of these reports, 75.3% had a positive test result. At 4.6%, the proportion of reports from medical practices is relatively small. The proportion of positive tests in this area is 45.6%. The proportion of severe illness in the sample as a whole is 4.1%. At 38.5%, the proportion is the highest for reports, which pertain to dental practices. However, this is based on few reports with positive test results (n = 13) only. The 11 deaths pertain to clinics (n = 3) and inpatient and outpatient care (n = 5), medical practices (n = 1), and care and counseling (n = 2). The mortality rate is 0.3% of all insured persons with a positive test result.

The most reported cases with a known test result pertained to nurses (63.9%). This is followed by physicians with a total of 1621 (15.0%) reports (Table 4). The average proportion of positive test results is 34.1%. Physicians were tested less often positive than nurses (24.3% versus 37.6%). Severe progressions are roughly twice as frequent among physicians as for the other occupational groups (8.1%, in comparison with an average of 4.1%). Seven nurses, two physicians and a social worker died. 

### 4.2. Safe Guarding Our HW during the COVID-19 Pandemic 2020—A Report from Malaysia

The Prime Minister of Malaysia, Muhyiddin Yassin, officially announced on 16 March, 2020, that the Malaysian Government would implement the Movement Control Order 2020 (MCO) as a preventive measure in response to the COVID-19 pandemic. The MCO, effective from 18 March, 2020, until 9 June, 2020, was implemented under the Prevention and Control of Infectious Diseases (1988) and the Police Act (1967). This MCO meant that Malaysians were subjected to “partial lockdown” whereby only necessary services were allowed to operate while others are to be withhold until further notice.

Since the increase of COVID-19 cases in Malaysia, beginning in late February, 2020, the local scenario experienced several clusters of COVID-19. As of 17 May, 2020, there were a total of 6894 laboratory confirmed cases. Out of these, 5571 (80.81%) have fully recovered and discharged from hospitals. A total of 1210 cases were treated in designated COVID-19 hospitals in Malaysia, with 13 patients in the intensive care unit, out of which 7 of these patients were on ventilators. There were 113 deaths recorded, with produced a case fatality rate of 1.64%.

The Ministry of Health, Malaysia reported that a total cumulative number of 224 health workers (HW) were diagnosed with COVID-19 as of 11 April, 2020. However, 80.0% of these HW cases were community acquired. They got infected due to joining mass gatherings such as religious congregations, as well as attending weddings. One example is that a total of 47 HW (including doctors, nurses, and medical assistants) were found positive after attending a wedding on 6 March, 2020, in Bangi, Selangor. Most were government HW and when they returned to their hospitals or healthcare centers, they then infected other colleagues. Hospitals in Kota Bharu Kelantan, Hospital Selayang in Selangor, and Hospital Teluk Intan in Perak, were among the badly affected centers. Since these HW had to be treated and quarantined for 14 days, the Ministry of Health had to deploy HW from other unaffected hospitals to overcome the sudden shortage of staff. There are currently 34,241 doctors serving in Malaysia (including 4970 medical specialists, 24,656 medical officers and 4545 house officers (interns). There are 13,904 Assistant Medical Officers and 65,709 nurses in the country [22].

Approximately 20% of the infected HW contracted the disease due to attending to patients in non-COVID-19 wards and intensive care units. The patients did not inform the HW that they had been in close contacts with COVID-19 patients. Therefore, the HW did not wear the full PPE when treating these patients. These are patients who came to the hospital with severe acute respiratory illness (SARI) and pneumonia. Therefore, during the early phase of the pandemic in Malaysia, these HW were exposed while waiting for the results of the throat/nasal swabs RT-PCR tests of such patients, which might take up to a few days due to overburden of our laboratory facilities. 

One positive observation is that until now, for those HW treating patients in designated COVID-19 hospitals, such as the General Hospital Kuala Lumpur and Hospital Sungai Buloh, Selangor, no HW were reported to have contracted the disease. These HW strictly follow the standard operation procedure (SOP) and the PPE instructions. 

Some HW from private hospitals contracted the disease because they did not wear the full PPE. Initially, in mid-March, 2020, the PPEs were not well distributed and some private hospitals and healthcare centers could not obtain the PPEs. However, the Ministry of Health arranged several series of meetings and discussions with the private hospitals consortium and the issue was quickly resolved. Currently, the supply of PPE is adequate and some local companies began to supply the Ministry of Health so as to reduce the dependency on imported PPE.

The Ministry of Health arranged screening for the HW and so far, 2000 HW (especially those in the frontlines and the high-risk zones) were screened. Out of these, 181 (9.1%) were reported to be positive. Among the strategies to protect the HW, the Ministry of Health implemented the assessment of needs and stockpiling of PPEs at all healthcare facilities to ensure adequate amounts, type, and quality of PPEs. Secondly, all SARI and pneumonia cases are treated as COVID-19 until proven otherwise. This means that all HC treating such cases must also wear full PPE, even though they are not working in designated COVID-19 wards or ICUs. Thirdly, the strengthening of the triaging system on assessment and isolation of patients at the Emergency Departments. When necessary, the patients will be provided with face masks. Fourth, all departments must ensure that HW treating COVID-19 patients must wear full PPE and adhere strictly to the SOP. Each HW must practice good hand hygiene and maintain social distancing. 

HW in Malaysia have reported feeling stress, tired, burnt out, and sad because some of them have not seen their families for months. They also reported sensing discrimination from the public as HW are perceived as “infected persons”, since they are working in hospitals or health centers. Ministry of Health has set up psychosocial support services for the front-liners and public. These services are being managed by psychiatrists, clinical psychologists, and counsellors to help HC and public to handle the depression, anxiety, insomnia, distress, and stigmatization that they faced during this COVID-19 pandemic. Most of us are hoping that the cases will continue to dwindle down, so that by 9 June, 2020, we will be able to proceed with our daily activities, amidst the “new-normal” life style. 

## 5. Discussion

COVID-19 has entirely changed the occupational occurrence of infections in Germany. In past years, roughly 10,000 claims of infection diseases suspected to be work-related were submitted to the BGW each year. Of these reported cases, 800 to 1000 per year were subject to mandatory reporting [16]. In the first four months of this year, the BGW has already received 12,038 suspected case reports due to SARS-CoV-2 and COVID-19, of which 4398 were subject to mandatory reporting. Eleven deaths and 151 severe illnesses demonstrate the particular vulnerability of health workers. Worldwide, 278 physicians died because of COVID-19 following an internet search on 15 April, 2020 [25]. This further highlights the burden of COVID-19 faced by HW.

The rate of severe illness is twice as high for physicians as for other workers (8.1% vs. 4.1%). This could be caused by testing and reporting behavior, which might differ from the other occupational groups. It is possible that doctors have themselves tested less often in cases of only mild symptoms. Assuming an equal risk for severe disease after infection and further assuming that the detection rate increases with the severity of the disease, underreporting of non-severe COVID-19 diseases in physicians might explain the higher rate of severe disease in the physicians infected. However, this assumption is contradicted by the observation that physicians were less likely to test positive then nurses. It may also be the case, however, that the higher rate of severe illness can be explained by an elevated type of exposure. The possibility that exposure to a higher virus load increases the risk of severe illness has been discussed in the literature [26]. The performance of bronchoscopies, intubations, or examinations of the head area could lead to higher levels of exposure in comparison with care activities.

The high proportion of mild progressions of COVID-19 (84.6%, Table 2) can possibly be explained by the fact that younger people exhibit severe progressions less frequently than older people [2]. However, this could also be explained by the fact that cases are discovered more actively for health workers in the context of contact tracings or occupational medical check-ups. As a result, cases are discovered which would not have been discovered otherwise due to a lack of symptoms. 

In the “Heinsberg Study”, 15% of the population was infected and the mortality rate among infected persons was 0.36% (95%, CI: 0.29–0.45) [27]. The median age of the study cohort was 53 years (min-max: 1–90 years). The mortality rate among infected patients calculated for health workers corresponds to the observation of the Heinsberg Study for both the RKI and the BGW data (0.2% to 0.5%). The average age of the hospital workers in the RKI data set is 41 years. For the other datasets, we do not have any age data. However, these pertain to cohorts of working subjects. For this reason, the average age is likely to be substantially below 60 years. 

In particular, the high mortality rate among infected workers demonstrates how important adequate infection protection of employees is in healthcare. There have been temporary bottlenecks for the procurement of the corresponding PPE. For this reason, the German Federal Institute for Occupational Safety and Health recommends that only non-medical mouth and nose protection be worn outside of a healthcare context (e.g., home-made fabric masks) [28]. Medical mouth and nose protection with corresponding certificates and FFP2/3 respirators should remain reserved for health workers. At this point, it should be noted that insurance coverage from accident insurance still applies in cases where sufficient PPE was unavailable or if it was not used. However, this is no reason to neglect the protection of workers against infection, since they could also infect patients. Of almost even greater importance is the fact that we cannot afford to lose health workers to illness because of insufficient protection from infection. There is also a moral obligation to mitigate workers’ fears for their own health and that of their relatives through the use of high-quality protection against infection. From the report about the situation in Malaysia, it becomes clear that an effective management of the pandemic and the appropriate supply and use of PPE can protect HW from infection. In Malaysia, those HW, who were infected at the workplace, did not use PPE sufficiently because they did not suspect the patients to be infectious. In addition, stress at work and the stigma HW have to deal with are testified in this report.

The two datasets from Germany presented here do have some limitations, which should be kept in mind. The data of the RKI, generated because of the reporting obligations of the Infection Protection Law, does not cover all health workers. For example, doctor offices and ambulatory care nurses are not assessed with their professions. In addition, RKI data do not allow to distinct work-related from private infections. The data of the accident insurance provider BGW, generated because of the reporting obligations of the Occupational Disease Ordinance (Berufskrankheitenverordnung), are likely subject of underreporting. A disease, which causes only mild symptoms for a few days, might not be reported, because the insurance is unlikely to grant any compensation for this disease. In addition, only half of the work-related infections in Germany pertain to workers covered by the BGW [29]. Therefore, the numbers of claims of an occupational disease because of COVID-19 should roughly be multiplied by two for an estimate of the effected health and social workers in Germany. The datasets of RKI and BGW do overlap. However, there is no way to verify to which extent they overlap. 

## 6. Conclusions

Following the presented data, 2192 cases of COVID-19 in health and welfare workers are confirmed as OD in Germany. The real number is likely to be more than twice as high and will increase, as not all claims are assessed yet. The example from Malaysia shows the additional stress and the stigma HW are facing during the pandemic. It also shows that systematic supply and use of PPE can effectively protect HW. Our level of knowledge regarding the spread of SARS-CoV-2 infections among HW remains relatively low. The gaps in our knowledge should be filled as quickly as possible by systematically analyzing the data from the occupational health examinations that are currently being carried out. 

## Figures and Tables

**Table 1 ijerph-17-04881-t001:** Cases of SARS-CoV-2 infections of employees from medical and care facilities.

Facility	Total		Hospitalized		Deceased		Recovered	
	N	% ^a^	N	% ^b^	N	% ^b^	N	% ^b^
Medical facility *	12,393	6.9	567	4.6	20	0.2	11,938	96.3
Care facility **	8935	5.0	382	4.3	46	0.5	8468	94.8
All reported COVID-19 cases in Germany	178,570	100.0	26.730 ^c^	17.8 ^c^	8257	4.6	161,200	90.3

* e.g., hospitals, medical practices, dialysis facilities, and ambulance services; ** e.g., nursing home, homeless shelters, facilities for housing asylum seekers, other mass accommodations, correctional facilities; ^a^ percentage of all reported cases in Germany; ^b^ percentage of the total of the respective group; ^c^ based on 150,287 COVID-19 cases with information on hospitalization status; according to RKI, 5/25/2020.

**Table 2 ijerph-17-04881-t002:** COVID-19 cases reported to the BGW, test results, and progressions of illnesses.

Claims	N	%
Reported cases	12,038	100.0
Mandatory reporting	4398	36.5
Cases with test result	10.835	90.0
Cases with positive test	3690	34.1 ^a^
Cases with finished evaluation	3067	83.1 ^b^
Cases confirmed as OD	2192	71.5 ^c^
Cases not confirmed as OD	875	28.5 ^c^
Cases with info on disease status	3038	82.3 ^b^
Cases with mild symptoms	2570	84.6 ^d^
Cases with severe symptoms	151	5.0 ^d^
Deaths	11	0.3 ^d^

^a^ Percentage of all reports with known test result; ^b^ percentage of all cases with a positive test; ^c^ percentage of all cases with finished evaluation; ^d^ percentage of all cases with information on disease status; OD = Occupational Disease.

**Table 3 ijerph-17-04881-t003:** COVID-19 cases reported to the accident insurance provider BGW by sector.

Sector	Cases with Known Test Result		Positive Test		Severe Progression		Deaths	
	N	% ^a^	N	% ^b^	N	% ^c^	N	% ^c^
Clinics	8110	74.9	1874	23.1	65	3.5	3	0.2
Inpatient and outpatient care	1762	16.3	1327	75.3	41	3.1	5	0.4
Medical practices	496	4.6	226	45.6	29	12.9	1	0.4
Dental practices	22	0.2	13	59.1	5	38.5	0	--
Therapeutic practices	122	1.1	38	31.1	3	7.9	0	--
Care and counseling	264	2.4	189	71.6	7	3.7	2	0.1
Pharmacy	6	0.05	2	33.3	1	50.0	0	--
Hair salons, beauty	2	0.02	0	--	0	--	0	--
Childcare	7	0.06	2	28.6	0	--	0	--
Other	44	0.4	19	43.2	0	--	0	--
Total	10,835	100.0	3690	34.1	151	4.1	11	0.3

^a^ Percentage of all cases with known test result; ^b^ percentage of all cases with test result within the sector; ^c^ percentage of all cases with positive test result within the sector.

**Table 4 ijerph-17-04881-t004:** COVID-19 cases reported to the accident insurance provider BGW broken down by activities/occupations.

Profession	Claims with Test Results		Positive Test		Severe Progression		Deaths	
	N	%^a^	N	%^b^	N	%^c^	N	%^c^
Physicians	1621	15.0	394	24.3	32	8.1	2	0.5
Nurses	6927	63.9	2605	37.6	90	3.5	7	0.3
Physiotherapist	214	2.0	60	28.0	2	3.3	0	--
Social worker	262	2.4	95	36.3	3	3.2	1	1.0
Other	1811	16.7	536	29.6	24	4.5	1	0.2
Total	10,835	100.0	3690	34.1	151	4.1	11	0.3

^a^ All cases with known test result; ^b^ all cases with test result within the sector; ^c^ all cases with positive test result.

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
