# Peer review of "COVID-19 among Health Workers in Germany and Malaysia"

_ijerph, 2020, doi:10.3390/ijerph17134881_

Round 1

Reviewer 1 Report

Thank you for the opportunity to revise this manuscript. The MS is interesting in the topic explored, but I believe that should be heavily restructured. It is my hope that Authors will find useful suggestions in the following points to review their manuscript.

General Considerations

  1. Even though the manuscript is overall understandable, at times the language used is odd and would benefit from being edited by someone whose first language is English. Also, the manuscript should be thoroughly edited for small mistakes and typos (e.g. line 18, line 48). (major)
  2. One thing that is not clear is what is the rationale behind considering Germany and Malaysia for this study. Why these two countries? How are they similar or different regarding the matter at hand? How this difference could have impacted the matter at hand? (major)
  3. What is the purpose of the study? It seems to me that it is merely a description and does not provide any insight regarding the COVID-19 situation for health workers. Authors should consider a new research design, formulating proper hypotheses and performing statistical analyses to verify them. (major)
  4. Most of the section 4.2 should be part of the Introduction rather than Results as it merely presents the situation in Malaysia. Also, in this section a lot of data and numbers are provided without an appropriate reference (e.g. line 225-229, 243, 252, 254, 271, etc.). An entire section can’t be based on personal experience or rumours (major).
  5. The discussion should be organized better, dividing the reflections by topics. Discussions are currently too confusing (major).

Detailed considerations

  1. Line 23. “Among workers”. Does this refer to workers in general or exclusively to health workers? This needs to be clarified. (minor)
  2. Line 22-25. You claim that COVID-19 suspected cases are 4,398 among workers and, later, you state that nurses represent most of these cases (n = 6,927). Something does not add up. (minor)
  3. In general. The first part of the Introduction is overall chaotic and does not clarify what the focus of the manuscript is going to be. This section should be restructured: first, Authors should present a general picture of COVID-19 and a more thorough literature search on this topic should be conducted. Then, Authors should focus on the literature regarding specifically health workers and COVID-19 and/or past pandemics.
  4. In general. It seems that the Introduction exclusively presents the evolution of the situation in Germany and ignores what is happening in Malaysia. Refer to point 2 and 4 of General Considerations.
  5. Line 37. To be clearer, Authors should present it as the “novel coronavirus”, since coronavirus alone refers to a family of viruses. (minor)
  6. At the end of the introduction, a more structured hypothesis must be made, or it should be clarified better because the study is only descriptive.

Author Response

Thank you for the opportunity to revise this manuscript. The MS is interesting in the topic explored, but I believe that should be heavily restructured. It is my hope that Authors will find useful suggestions in the following points to review their manuscript.

General Considerations

1.      Even though the manuscript is overall understandable, at times the language used is odd and would benefit from being edited by someone whose first language is English. Also, the manuscript should be thoroughly edited for small mistakes and typos (e.g. line 18, line 48). (major)

Author’s response: The paragraph that seem odd in a scientific paper is deleted, following the suggestions of the other reviewers. The sentence in line 48 was deleted. The paper was proof read by a native speaker.

2.      One thing that is not clear is what is the rationale behind considering Germany and Malaysia for this study. Why these two countries? How are they similar or different regarding the matter at hand? How this difference could have impacted the matter at hand? (major)

Author’s reply: We invited other colleagues to contribute. However, Malaysia was the only country to answer to our appeal. Trying to get as much evidence published about Covid-19 in HW in different countries justifies compromises with scientific writing. Don’t you agree? Now we explain why we combine a report from Malaysia with one from Germany.

3.      What is the purpose of the study? It seems to me that it is merely a description and does not provide any insight regarding the COVID-19 situation for health workers. Authors should consider a new research design, formulating proper hypotheses and performing statistical analyses to verify them. (major)

Authors reply: Yes, you are right. It is a descriptive study. Did you know about the numbers in Germany and Malaysia? I guess no. Now, you got an idea. That is the rationale behind this manuscript. We rephrased the objectives of the study: ‘In this descriptive study, we report the cases of SARS-CoV-2 infections and COVID-19 illnesses in HW in Germany. In addition a report on the COVID-19 situation for HW in Malaysia is given.’

4.      Most of the section 4.2 should be part of the Introduction rather than Results as it merely presents the situation in Malaysia. Also, in this section a lot of data and numbers are provided without an appropriate reference (e.g. line 225-229, 243, 252, 254, 271, etc.). An entire section can’t be based on personal experience or rumours (major).

Author’s response: you are right. However here we are not talking about a study but a personal report concerning one country. You can’t ask an interview partner for references.

5.      The discussion should be organized better, dividing the reflections by topics. Discussions are currently too confusing (major).

Author’s response: Thank you for your comment. We rearranged the discussion by eliminating some mistakes.

Detailed considerations

1.      Line 23. “Among workers”. Does this refer to workers in general or exclusively to health workers? This needs to be clarified. (minor)

Author’s response: It is among health and social workers. We amended this.

2.      Line 22-25. You claim that COVID-19 suspected cases are 4,398 among workers and, later, you state that nurses represent most of these cases (n = 6,927). Something does not add up. (minor)

Author’s response. We agree this is confusing. We talk about confirmed infections and about claims filled. A claim can be filled without an infection being confirmed. Now we introduced: ‘regardless of being a confirmed infection’.

3.      In general. The first part of the Introduction is overall chaotic and does not clarify what the focus of the manuscript is going to be. This section should be restructured: first, Authors should present a general picture of COVID-19 and a more thorough literature search on this topic should be conducted. Then, Authors should focus on the literature regarding specifically health workers and COVID-19 and/or past pandemics.

Author’s response: We changed the introduction accordingly and we integrated more literature. See comment of reviewer 2 and our answer.

4.      In general. It seems that the Introduction exclusively presents the evolution of the situation in Germany and ignores what is happening in Malaysia. Refer to point 2 and 4 of General Considerations.

Author’s response: we deleted the prose referring to Germany. This should have solved the problem.

5.      Line 37. To be clearer, Authors should present it as the “novel coronavirus”, since coronavirus alone refers to a family of viruses. (minor)

Author’s response: Yes, you are right. However, this sentence was deleted.

6.      At the end of the introduction, a more structured hypothesis must be made, or it should be clarified better because the study is only descriptive.

Author’s response: Yes, you are right. It is a descriptive study. We think it is important to publish the number of HW affected as well in Germany as in Malaysia or in any other country. Now we start the objective with ‘In this descriptive study’

Reviewer 2 Report

Important, good, interesting and topical contribution to the special burden of health workers in the context of the Covid 19 pandemic

Comments/questions/recommendations

Basic question: why was only a comparison with Malaysia presented and not including results from other countries, eg [1–6]

Introduction, in particular Line 37-64: I find the introduction a bit long, it goes into detail about the actually generally known pandemic development (internationally and in Germany). It could be compressed a bit.

Chapter 4.2: the results from Malaysia could also be summarized here in a table, if necessary also in comparison to the (additionally) other international studies mentioned

Line 261: possibly add the percentage value “Out of these, 181 (9,1%) were reported…”

In the introduction and the discussion, reference could also be made to Holmes et al to the “Multidisciplinary research priorities for the COVID-19 pandemic”[7]

Translated by www.DeepL.com/Translator (free version)

Literatur

1      Wang S, Xie L, Xu Y et al. Sleep disturbances among medical workers during the outbreak of COVID-2019. Occup Med (Lond) 2020; DOI: 10.1093/occmed/kqaa074

2      Ranka S, Quigley J, Hussain T. Behaviour of occupational health services during the COVID-19 pandemic. Occup Med (Lond) 2020; DOI: 10.1093/occmed/kqaa085

3      Ing EB, Xu QA, Salimi A et al. Physician deaths from corona virus (COVID-19) disease. Occup Med (Lond) 2020; DOI: 10.1093/occmed/kqaa088

4      Williamson V, Murphy D, Greenberg N. COVID-19 and experiences of moral injury in front-line key workers. Occup Med (Lond) 2020; DOI: 10.1093/occmed/kqaa052

5      Lai X, Wang M, Qin C et al. Coronavirus Disease 2019 (COVID-2019) Infection Among Health Care Workers and Implications for Prevention Measures in a Tertiary Hospital in Wuhan, China. JAMA Netw Open 2020; 3 (5): e209666; DOI: 10.1001/jamanetworkopen.2020.9666

6      Koh D. Occupational risks for COVID-19 infection. Occup Med (Lond) 2020; 70 (1): 3–5; DOI: 10.1093/occmed/kqaa036

7      Holmes EA, O'Connor RC, Perry VH et al. Multidisciplinary research priorities for the COVID-19 pandemic. A call for action for mental health science. The Lancet Psychiatry 2020; DOI: 10.1016/S2215-0366(20)30168-1

Author Response

Important, good, interesting and topical contribution to the special burden of health workers in the context of the Covid 19 pandemic

Comments/questions/recommendations

Basic question: why was only a comparison with Malaysia presented and not including results from other countries, eg [1–6]

Author’s response: We did not intend to present a comparison between Germany and Malaysia. We intend to render as much data as possible accessible for the public. Malaysia was the only country to provide a report upon our invitation sent to all participants of the OHHW 2019 conference in Hamburg. Therefore, our paper is not a comparison study.

Introduction, in particular Line 37-64: I find the introduction a bit long, it goes into detail about the actually generally known pandemic development (internationally and in Germany). It could be compressed a bit.

Author’s response: This is in line with the comments of the other reviewers. We deleted the first paragraph.

Chapter 4.2: the results from Malaysia could also be summarized here in a table, if necessary also in comparison to the (additionally) other international studies mentioned

Author’s response: As mentioned before these are case reports from two countries. It is not a review of COVID-19 in HW worldwide. This would be an important task to be performed in the next months. Therefore, we prefer not to compile an additional table.

Line 261: possibly add the percentage value “Out of these, 181 (9,1%) were reported…”

Author’s response: Thank you for your comment. We added this.

In the introduction and the discussion, reference could also be made to Holmes et al to the “Multidisciplinary research priorities for the COVID-19 pandemic”[7]

Translated by www.DeepL.com/Translator (free version)

Literatur

1      Wang S, Xie L, Xu Y et al. Sleep disturbances among medical workers during the outbreak of COVID-2019. Occup Med (Lond) 2020; DOI: 10.1093/occmed/kqaa074

2      Ranka S, Quigley J, Hussain T. Behaviour of occupational health services during the COVID-19 pandemic. Occup Med (Lond) 2020; DOI: 10.1093/occmed/kqaa085

Survey on about 60 OSH specialists. interesting but not cited

3      Ing EB, Xu QA, Salimi A et al. Physician deaths from corona virus (COVID-19) disease. Occup Med (Lond) 2020; DOI: 10.1093/occmed/kqaa088

4      Williamson V, Murphy D, Greenberg N. COVID-19 and experiences of moral injury in front-line key workers. Occup Med (Lond) 2020; DOI: 10.1093/occmed/kqaa052

Editorial not cited

5      Lai X, Wang M, Qin C et al. Coronavirus Disease 2019 (COVID-2019) Infection Among Health Care Workers and Implications for Prevention Measures in a Tertiary Hospital in Wuhan, China. JAMA Netw Open 2020; 3 (5): e209666; DOI: 10.1001/jamanetworkopen.2020.9666

6      Koh D. Occupational risks for COVID-19 infection. Occup Med (Lond) 2020; 70 (1): 3–5; DOI: 10.1093/occmed/kqaa036

7      Holmes EA, O'Connor RC, Perry VH et al. Multidisciplinary research priorities for the COVID-19 pandemic. A call for action for mental health science. The Lancet Psychiatry 2020; DOI: 10.1016/S2215-0366(20)30168-1

Authors response: Thank you for the literature. With two exception (no. 2 and 7) the literature is now integrated in the paper.

Reviewer 3 Report

General comments

The reported data are interesting and relevant in the field of public health. However, I see no connection between the reports from Germany and Malaysia. If there would be striking differences and a comparison of the data, it would have been understandable but in the recent form its 2 independent reports.

Specific Comments:

Line 37-51: This is too illustrative for a scientific publication. Its really good written but it fits not at all as introduction to this topic.

Line 53: A reference is missing

Line 64: A reference is missing

Line 110: A reference is missing for the 29%

Line 162: Which groups?

Line 168: I think it must be SARS-CoV-2 infections in the header.

Line 180: The sentence starts with a number.

Line 248: You mean RT-PCR?

Lines 290/291: if underreported why twice as high (see line 288)?

The tables are not referenced in the text.

Author Response

General comments

The reported data are interesting and relevant in the field of public health. However, I see no connection between the reports from Germany and Malaysia. If there would be striking differences and a comparison of the data, it would have been understandable but in the recent form its 2 independent reports.

Authors response: we agree, these are two reports in one paper. We invited others to give brief reports. Besides of the report from Malaysia we received a report from India. However, this report was not included but submitted as a short communication. I think it is charming to have two reports from two attendees of the OHHW 2019 conference in Hamburg in the conference proceedings. This shows that the conference facilitated cooperations. Now we explain this in the introduction.

Specific Comments:

Line 37-51: This is too illustrative for a scientific publication. Its really good written but it fits not at all as introduction to this topic.

Authors response: Thank you for pointing this out. You are in agreement with the other two reviewers. We deleted this paragraph.

Line 53: A reference is missing

Authors response: Thank you for pointing this out. A reference was added: Koh D: Occupational risks for COVID-19 infection. Occup Med (Lond) 2020, 70 (1), 3–5

Line 64: A reference is missing

Authors response: Thank you for pointing this out. A reference was added: Lai X, Wang M, Qin C et al. Coronavirus Disease 2019 (COVID-2019) Infection Among Health Care Workers and Implications for Prevention Measures in a Tertiary Hospital in Wuhan, China. JAMA Netw Open 2020; 3 (5): e209666; DOI: 10.1001/jamanetworkopen.2020.9666

Line 110: A reference is missing for the 29%

Authors response: Thank you for pointing this out. It was corrected: Ref RKI 2020, former Ref 1 now Ref 2

Line 162: Which groups?

Authors response: Thank you for pointing this out. We added “the medical facility and the care facility group”

Line 168: I think it must be SARS-CoV-2 infections in the header.

Authors response: Thank you for pointing this out. This was corrected

Line 180: The sentence starts with a number.

Authors response: Thank you for pointing this out. We rearranged the sentence.

Line 248: You mean RT-PCR?

Authors response: You are right. The typo was corrected

Lines 290/291: if underreported why twice as high (see line 288)?

Authors response: Thank you for pointing this out. We tried to explain our reasoning better. Now we write: Assuming an equal risk for severe disease after infection and further assuming that the detection rate increase with the severity of the disease, underreporting of non-severe Covid-19 diseases in physicians might explain the higher rate of severe disease in the physicians infected. 

The tables are not referenced in the text.

Authors response: Thank you for pointing this out. It was corrected.

Round 2

Reviewer 1 Report

The authors answered the considerations of the first revision of the MS. In particular, the rationale of the study and the research design are clearer.

Author Response

Reviewer 1 Second response

The authors answered the considerations of the first revision of the MS. In particular, the rationale of the study and the research design are clearer.

Author’s response: Thank you for your comment

Reviewer 3 Report

Line 279: Please add the date for which this number is valid - this makes the understanding easier and avoids missinterpretions after the pandemic.

Table 2: The last 3 lines are higher and the numbers are not aligned (format issue).

The reference section must be corrected. Many references are not present (reference manager issue).

Author Response

Reviewer 3 Second response

Line 279: Please add the date for which this number is valid - this makes the understanding easier and avoids missinterpretions after the pandemic.

Author’s response: Thank you for pointing this out. We added April 15th 2020 as the date of the internet research

Table 2: The last 3 lines are higher and the numbers are not aligned (format issue).

Author’s response: This was corrected.

The reference section must be corrected. Many references are not present (reference manager issue).

Author’s response: Thank you for pointing this out. We carefully checked the reference list and the references in the paper. Now, we present any reference listed and we avoid something like [2-9].

Thank you very much for critically reading the paper for a second time.